# Universal Prime Editing Therapeutic Strategy for RyR1-Related Myopathies: A Protective Mutation Rescues Leaky RyR1 Channel

**DOI:** 10.3390/ijms26072835

**Published:** 2025-03-21

**Authors:** Kelly Godbout, Mathieu Dugas, Steven R. Reiken, Sina Ramezani, Alexia Falle, Joël Rousseau, Anetta E. Wronska, Gabriel Lamothe, Geoffrey Canet, Yaoyao Lu, Emmanuel Planel, Andrew R. Marks, Jacques P. Tremblay

**Affiliations:** 1Molecular Medicine Department, Laval University, Quebec, QC G1V 0A6, Canada; mathieu.dugas@crchudequebec.ulaval.ca (M.D.); sina.ramezani@crchudequebec.ulaval.ca (S.R.); alexia.falle@crchudequebec.ulaval.ca (A.F.); gabriel.lamothe@crchudequebec.ulaval.ca (G.L.); geoffrey.canet.1@ulaval.ca (G.C.); yaoyao.lu@crchudequebec.ulaval.ca (Y.L.); emmanuel.planel@fmed.ulaval.ca (E.P.); 2CHU de Québec Research Center-Laval University, Quebec, QC G1V 4G2, Canada; joel.rousseau@crchudequebec.ulaval.ca; 3Department of Physiology and Cellular Biophysics, Center for Molecular Cardiology, Columbia University Vagelos College of Physicians and Surgeons, New York, NY 10032, USA; sr372@cumc.columbia.edu (S.R.R.); aew2116@cumc.columbia.edu (A.E.W.); arm42@cumc.columbia.edu (A.R.M.)

**Keywords:** RyR1-RM, prime editing, *RYR1* gene, CRISPR/Cas9, protective mutation, calcium channel, calcium leak

## Abstract

RyR1-related myopathies (RyR1-RMs) include a wide range of genetic disorders that result from mutations in the *RYR1* gene. Pathogenic variants lead to defective intracellular calcium homeostasis and muscle dysfunction. Fixing intracellular calcium leaks by stabilizing the RyR1 calcium channel has been identified as a promising therapeutic target. Gene therapy via prime editing also holds great promise as it can cure diseases by correcting genetic mutations. However, as more than 700 variants have been identified in the *RYR1* gene, a universal treatment would be a more suitable solution for patients. Our investigation into the RyR1-S2843A mutation has yielded promising results. Using a calcium leak assay, we determined that the S2843A mutation was protective when combined with pathogenic mutations and significantly reduced the Ca^2+^ leak of the RyR1 channel. Our study demonstrated that prime editing can efficiently introduce the protective S2843A mutation. In vitro experiments using the RNA electroporation of the prime editing components in human myoblasts achieved a 31% introduction of this mutation. This article lays the foundation for a new therapeutic approach for RyR1-RM, where a unique once-in-a-lifetime prime editing treatment could potentially be universally applied to all patients with a leaky RyR1 channel.

## 1. Introduction

Ryanodine receptor 1-related myopathies (RyR1-RMs) are the most common non-dystrophic congenital myopathies, with an estimated prevalence of 1 in 90,000 in the United States [1]. These disorders encompass a spectrum of conditions, including central core disease, multi-minicore disease, malignant hyperthermia, centronuclear myopathy, congenital fiber-type disproportion, and exertional rhabdomyolysis [2]. These genetic disorders are inherited in either an autosomal dominant or recessive pattern and are characterized by symptoms such as muscle cramps, fatigue, heat-related illnesses, respiratory impairments, post-exercise muscle pains, and malignant hyperthermia [3]. RyR1-RMs arise from mutations in the *RYR1* gene, which encodes the ryanodine receptor 1 (RyR1), a calcium release channel that forms homo-tetramers embedded in the sarcoplasmic reticulum (SR) of skeletal muscle fibers [4]. Mutations in *RYR1* disrupt calcium homeostasis, leading to pathological SR calcium leak, the depletion of SR calcium stores, mitochondrial calcium overload, and oxidative stress, which collectively contribute to muscle weakness and tissue damage [5,6,7].

Stabilizing RyR1 function has emerged as a promising therapeutic strategy for RyR1-RMs [5,6,8]. Calstabin binds to RyR1 and stabilizes its closed conformation [9], preventing pathological calcium leak [10]. Ser2843 has been identified as a major regulatory phosphorylation site implicated in RyR1 channel modulation. When RyR1 is hyperphosphorylated by protein kinase A (PKA) at the Ser2843 site, which occurs during stress or exercise, it promotes calstabin dissociation [11,12]. As a result, the stability of the closed state of the RyR1 channel is reduced, resulting in a pathological leak of Ca^2+^ [13] (Figure 1a), which impairs contractility and damages the muscle [8]. Rycal compounds (RyR calcium-release channel stabilizers), which prevent calstabin dissociation, have been shown to reduce calcium leak and improve muscle function in preclinical models and early-phase clinical trials [14,15], further supporting the therapeutic potential of RyR channel stabilizers [6].

Given the importance of Ser2843 phosphorylation in RyR1 regulation, we hypothesized that introducing a phosphorylation-deficient S2843A mutation into *RYR1* carrying pathogenic mutations could mimic the effects of Rycal therapy. The S2843A mutation (where a Serine (S) is replaced with an Alanine (A)) was initially generated to investigate the role and mechanism of phosphorylation in RyR1 regulation [12]. This mutation has been shown to remove an important phosphorylation site, enhance calstabin binding, and reduce RyR1 open probability (Po) [16,17] (Figure 1b). To test this hypothesis, we utilized prime editing [18,19,20,21], a precise gene-editing technology derived from CRISPR/Cas9, to introduce the S2843A mutation as a gene therapy. Prime editing offers a flexible approach, as it can introduce insertions, deletions, and all types of substitutions, while avoiding double-strand breaks. This approach has already been applied successfully to correct point mutations in *RYR1* [22]. However, one major drawback of this point mutation correction strategy is that it must be optimized for each mutation, limiting its feasibility for treating rare diseases with diverse genetic variants. RyR1-RMs alone encompass over 700 known mutations [4], making personalized gene therapy impractical due to cost and regulatory constraints. A mutation such as S2843A, which broadly stabilizes RyR1 function, could perhaps represent a “universal” therapeutic strategy applicable to a wide range of RyR1-RM patients.

To assess this strategy, we first evaluated the protective effect of the S2843A mutation in a calcium leak assay. We then designed and optimized prime editing components to introduce this mutation into cultured human myoblasts, achieving successful editing in 31% of *RYR1* alleles.

## 2. Results

### 2.1. The S2843A Mutation Rescued the RyR1 Channel Leak Caused by Pathogenic Mutations in the RYR1 Gene

We previously reported pathological Ca^2+^ leak from SR microsomes isolated from patients with RyR1-related myopathies [5]. Recombinant RyR1 channels harboring the R1667C and L714del mutations exhibited a “leaky” phenotype characterized by increased channel Po when reconstituted in a planar lipid bilayer. These mutations, located near the calstabin1 binding site, impaired calstabin1 binding and destabilized the closed state of the channel [5]. Microsomal Ca^2+^ leak was significantly elevated in channels with the R1667C and L714del mutations compared to the control (Figure 2).

To investigate the potential therapeutic impact of modifying RyR1 PKA phosphorylation, we introduced the S2843A mutation into RyR1 channels containing the pathogenic mutations. Substituting a serine at position 2843 with alanine renders the site non-phosphorylatable by PKA [12,13,23]. Remarkably, introducing the S2843A mutation into these leaky mutant channels significantly reduced the channel Po, effectively reducing the Ca^2+^ leak (Figure 2). We then compared this approach to the Rycal S107 drug. Experimental results showed that both the S2843A mutation and the S107 drug significantly reduced Ca^2+^ leak in mutant RyR1 channels with the two pathogenic mutations but did not completely restore function to that of the WT channels. Five independent trials (Appendix A) confirmed that the S2843A mutation reduced Ca^2+^ leak to a similar extent as S107 (Figure 2). These findings confirm our hypothesis and provide a strong rationale for developing a prime editing system to introduce the S2843A mutation as a one-time therapeutic intervention for RyR1-related myopathies.

### 2.2. Prime Editing Design to Introduce the S2843A Mutation into the RYR1 Gene

Three different spacers were tested to find the best prime editing design to introduce the S2843A protective mutation (Figure 3a–c). All constructs were tested by plasmid DNA lipofection in HEK293T cells. Engineered pegRNAs (epegRNAs) were used. They are an improved and more suitable version containing a secondary structure, in this case, a pseudoknot in the 3′ end, which stabilizes the RNA and limits its degradation with exonucleases [24]. Spacer 1 used a classical PAM (NGG), but the targeted nucleotide was at +13 from the cut site. Nine different combinations of RTT and PBS lengths were tested (15, 17, or 19 nt for the RTT and 10, 14, or 17 nt for the PBS). The PAM was not disrupted in the configuration of those epegRNAs. The epegRNA with Spacer 1 had a nsgRNA at +45. Unfortunately, none of those combinations tested for this spacer worked. We believe this was due to the long stretch of thymines (Ts) in the RTT-PBS sequence, which comprised seven Ts in a row. Given that there is a U6 promoter in the plasmid, such a long stretch of Ts would generate a stop signal preventing the epegRNA transcription [25,26,27]. Without enough epegRNA, the intended edit will not be incorporated efficiently.

Spacer 2 used an NGAN PAM. The intended substitution was at +6 from the cut site and was thus disrupting the PAM at the same time. Nine different combinations of RTT and PBS lengths were tested (12, 14, or 16 nt for the RTT and 10, 13, or 16 nt for the PBS). Firstly, PE-VQR was used for those epegRNAs [28]. The nsgRNA used was at +53. Unfortunately, none of those combinations tested for this spacer outperformed the negative control. It should be noted that the epegRNA with Spacer 2 used an NGA PAM and, as such, needs a prime editor with the VQR mutations in Cas9 [28]. This prime editor is supposed to be good for the NGAN PAM but is inefficient for the NGAT PAM, which was present in our sequence. Thus, those epegRNAs were also tested with a form of PE that recognizes NG PAMs (PE-NG). Although the editing rate was better than with PE-VQR, it was still lower than with the third version of the spacer.

Spacer 3 used a classical PAM (NGG), but the targeted edit was at +16 from the cut site. This configuration was designed using the PRIDICT program [29,30]. An RTT of 24 nt and a PBS of 12 nt were used. The PAM was disrupted in the configuration of this epegRNA (5′-GGG-3′ to 5′-GAG-3′). With the PAM being out of frame and on the non-coding strand, its disruption did not change the amino acid. The epegRNA with Spacer 3 was combined with an nsgRNA at +50. In spite of this configuration having the intended edit far from the cut site (+16), the S2843A mutation was introduced into 18% of HEK293T cells (Figure 3d). Spacer 3 was therefore used for the subsequent experiments.

To better represent the cell type in which the therapy would take place, the prime editing components were subsequently tested in human myoblasts. Plasmids coding for the epegRNA with the Spacer 3 configuration, its nsgRNA, and the PE were electroporated in wildtype (WT) myoblasts. The PAM was disrupted in 29% of *RYR1* genes, and the S2843A mutation was introduced in 11% of *RYR1* genes (Figure 3e). Since an 11% introduction may not be sufficient to achieve phenotypic improvement, the strategy has been further optimized.

### 2.3. Optimization of the epegRNA by Adding Silent Mutations Between the PAM and the +16 Position of the RTT

Several combinations of silent (same-sense) mutations between the PAM and the S2843A mutation were tested (Figure 4). Those constructs were electroporated in myoblasts. Figure 5 shows that none of the new configurations with additional silent mutations significantly improved prime editing efficiency. For PAM disruption, the original (+) epegRNA reached 18.5% efficiency. Some configurations, like epegRNA A, B, F, and G, showed slightly higher efficiencies at 25%, 21.5%, 18.5%, and 21%, respectively, but were not significantly better than the original one. Other variations, like epegRNA C and I, disrupted the PAM in only 12.75% of the *RYR1* genes, which was significantly lower than the original with Spacer 3 (+) and some other combinations (Figure 5b).

The same pattern was similarly observed when introducing the S2843A at +16. None of the new configurations with additional silent mutations significantly improved the efficiency of introducing the S2843A mutation. The original (+) epegRNA introduced this mutation at +16 in 7.5% of myoblasts. Some configurations, like epegRNA A, B, and G, obtained slightly superior efficiencies at 10%, 7.8%, and 8.7%, respectively, but were not significantly higher than the original. Other variations, like epegRNA C and I, introduced the S2843A mutation at only 4%, which was significantly lower than the original and some other epegRNAs (Figure 5c). It is also interesting to note that none of the substitutions (A → G/C/T) at +14 significantly affected the overall editing efficiency (Figure 5b,c). Interestingly, the nature of silent mutations did not appear to affect efficiency, as demonstrated by similar performance across epegRNAs with varied nucleotides at position +14. This suggests that the spatial configuration of silent edits plays a more significant role than their biochemical properties.

Previous studies have also shown that providing adjacent edits increases the prime editing efficiency by helping the desired edit escape the mismatch repair system [31]. Unfortunately, no silent mutations were possible at position +15, which may explain why our silent mutations did not significantly improve the prime editing efficiency (Figure 5). Since adding silent mutations did not improve editing rates, the original version without additional silent mutations was used for subsequent experiments.

### 2.4. The +11 Silent Mutation in the RTT Decreased Prime Editing Efficiency

EpegRNAs introducing the A>G silent mutation at +11 from the cut site demonstrated significantly reduced editing efficiency compared to those without this mutation (*p*-value < 0.0001). EpegRNA lacking the +11 silent mutation (epegRNAs +, A, B, F, and G) achieved an average PAM disruption rate of 21% (Figure 6a) and introduced the S2843A mutation at +16 position at an average rate of 7.9% (Figure 6b). In contrast, epegRNAs containing the +11 silent mutation (epegRNAs C, D, E, H, I, and J) showed reduced performance, with an average PAM disruption rate of 14% (Figure 6a) and a lower S2843A editing rate of 4.5% (Figure 6b). Although this configuration did not enhance our editing rates, this unexpected finding suggests that specific nucleotide positions in the RTT region may influence editing outcomes, independent of secondary structure predictions (Appendix A).

### 2.5. Assessing PE Variants to Improve Editing Efficiency

Having found a number of suitable epegRNAs (B, F, and G), the next step was to determine their efficacy using various generations of prime editors. PE3 contains an additional nicking single-guide RNA (nsgRNA) [32] designed to cut the unedited strand close to the editing site. This single-strand cut improves the editing efficiency by increasing the chances of the edited strand being preserved over the non-edited strand. The editing efficiency of PE3 is generally twice that achieved with PE2 [22]. As some specific DNA mismatch repair (MMR) proteins strongly reduce the efficiency of prime editing, Chen et al. [31] designed PE5. PE5 is based on PE3 but includes the co-expression of the MLH1dn gene, which encodes a mismatch repair inhibitory protein. Doman et al. [33] used phage-assisted evolution to choose advantageous mutations in the PE and a smaller version of the RT domain, generating the PE6 version of prime editing. The prime editing components were electroporated into myoblasts.

Here, we found that later variants of PE, including PE5 [31], PE5max [31], and PE6 [33], did not yield significantly higher PAM disruption efficiencies (Figure 7a). In contrast, the PE5 strategy, which suppresses specific MMR components, did not significantly improve the efficiency over that of PE3. PE5max was found to cause a significant reduction in editing with epegRNA B compared with PE3. However, PE6, a more compact version of PE, resulted in significantly higher efficiencies for inserting the S2843A mutation at position +16 for all epegRNAs tested compared to PE3. PE6 reached 11.5% editing for epegRNA B and G and 10% for epegRNA F, compared with 8%, 6.5%, and 7% editing obtained using PE3 (Figure 7b). The compact size of PE6 makes it particularly suited for delivery in scenarios with vector size limitations, further enhancing its therapeutic potential. Thus, the PE6 strategy was used for subsequent experiments.

### 2.6. RNA Delivery Improved Prime Editing Efficiencies

Delivering prime editing components as RNA rather than plasmid DNA has been previously demonstrated to enhance editing efficiency [22]. In this study, the original epegRNA with Spacer 3 was delivered to myoblasts via electroporation in both DNA and RNA formats. The PAM was not disrupted in the epegRNA configuration. By preserving the PAM, we achieved a balance between a high editing efficiency and the ability to target downstream mutations. Using RNA ratios and quantities optimized in our prior work [22], the S2843A was successfully introduced into 22% of *RYR1* alleles in myoblasts (Figure 8). As anticipated, reducing the delivered RNA quantity by half resulted in a proportional decrease in editing efficiency, yielding 11.5% editing. Interestingly, doubling the RNA quantity increased the editing efficiency, achieving a 31% introduction of the S2843A mutation at its +16 position.

### 2.7. Off-Target Analysis

An off-target analysis was performed for this spacer using IDT’s online tool [34]. Most off-target sequences identified were in non-coding regions and contained at least three or four mismatches/bulges. However, one off-target site was in the intron of the *SMPD4* gene, and only had two mismatches with the epegRNA’ spacer at positions 10 and 15. In addition, an off-target in the coding sequence of the *EFNA1* gene was also found, with two mismatches (at positions 10 and 15) and one bulge (at position 16). Thus, these sites in the *SMPD4* and *EFNA1* genes were amplified and analyzed. No off-target events were detected at those sites by Sanger sequencing. This observation aligns with what has been reported in the literature, where prime editing is shown to be highly specific, with no detectable off-target effects in several studies [35,36,37,38,39]. These results further support the safety of this therapeutic approach.

## 3. Discussion

This study demonstrates the potential of introducing the S2843A mutation into the *RYR1* gene via prime editing as a therapeutic approach for RyR1-RM. Our findings revealed that S2843A effectively stabilizes leaky RyR1 channels, offering a mechanism of action comparable to pharmacological treatments like Rycals. The potential therapeutic scope of the S2843A mutation could match that of Rycal treatments, which have proven effective across various RyR1 mutations causing leaky channels [5]. This suggests that S2843A could benefit a broad spectrum of RyR1-RM patients, potentially serving as a one-size-fits-all solution for stabilizing leaky RyR1 channels. Unlike pharmacological treatments, which typically require lifelong adherence, gene therapy offers the potential for a one-time, durable solution. By expanding the therapeutic landscape, both approaches contribute to meeting the diverse needs of RyR1-RM patients.

Evidence from a previous study supports the safety of the S2843A mutation. In a murine model, Andersson et al. [23] showed that its equivalent (S2844A) had no adverse effects on body weight, skeletal muscle histology, or force-generating capacity. These findings reinforce its potential as a therapeutic intervention.

Prime editing also offers a promising avenue for correcting point mutations within the *RYR1* gene [22]. However, personalized prime editing approaches require tailored epegRNAs for each specific mutation, which complicates regulatory approval and increases treatment costs. As more than 700 mutations in the *RYR1* gene have been identified [4], a universal approach like the one proposed here could streamline regulatory approval processes and lower the financial burden associated with individualized treatments.

Several technical hurdles must be overcome before S2843A prime editing can be translated into clinical applications. Ensuring that the safety of prime editing is a primary concern, particularly regarding potential off-target effects. Whole-genome sequencing and chromosome mapping will be essential to evaluate and mitigate these risks comprehensively. In parallel, optimizing delivery methods is crucial for achieving high in vivo efficiency. Many possibilities are now being investigated to find the most suitable and safe delivery methods, like dual adeno-associated viruses [20], virus-like particles [40], and lipid nanoparticles [41,42].

Despite these challenges, prime editing continues to advance as a powerful therapeutic tool. Expanding Cas9 libraries with broader PAM recognition could improve editing efficiency by nicking the DNA closer to the intended edit. Determining the threshold editing rate required for therapeutic benefits in RyR1-RM patients will also be critical. The present study lays the groundwork for a novel therapeutic strategy for RyR1-RM. The introduction of the S2843A mutation offers the potential for a once-in-a-lifetime prime editing treatment applicable to a wide range of patients with leaky RyR1 channels, paving the way for a cost-effective and accessible solution to a complex genetic disorder.

## 4. Materials and Methods

### 4.1. Prime Editing Technology

The prime editor (PE) consists of a Cas9 nickase protein fused to a reverse transcriptase and a prime editing guide (pegRNA) [32]. From 5′ to 3′, the pegRNA is composed of the spacer, the scaffold, the reverse transcriptase template (RTT), and the primer binding site (PBS). The spacer is a sequence complementary to 20 nucleotides (nt) in the genome, which guides nCas9 to the desired site. The pegRNA is designed so that a protospacer adjacent motif (PAM) is located 3′ from the sequence identical to the spacer. In the case of the Cas9 most frequently used in prime editing [i.e., *Streptococcus pyogenes*-derived Cas9 (SpCas9)], the PAM is the 5′-NGG-3′ sequence. The PBS sequence is complementary to a sequence in 5′ of the SpCas9 cut site. The RTT serves as a template for the reverse transcriptase to regenerate the sequence 3′ of the cut, thus inserting a different nucleotide sequence into the DNA. One of the great advantages of this technology is that it creates only a single-strand break in the DNA, unlike the conventional CRISPR/Cas9 system, which induces a double-strand break, thus reducing the damage to the DNA and risks associated with double-strand breaks [21].

### 4.2. Plasmids

The pCMV-PE2 (Prime editor), pU6-tevopreq1-GG-acceptor plasmids (epegRNA), pCMV-PE2-NG, and pCMV-PE6a were acquired from AddGene (Cambridge, MA, USA) (respectively, AddGene plasmids #132775, #174038, #159977, and #207851). Cloning in the pU6-tevopreq1-GG-acceptor plasmid followed the protocol described by Anzalone et al. [32] to create the epegRNA plasmids. Our team customized plasmid #174038 to incorporate a second U6 promoter and a cloning site to insert the nsgRNA for the PE3 method (subsequently referred to as epegRNA-nsgRNA). Oligonucleotides necessary for constructing epegRNAs and nsgRNAs were procured from IDT Inc. (Coralville, IO, USA). The spacer, PBS, RTT, and nsgRNA sequences are detailed in Figure 4 and Appendix A.

### 4.3. Cell Line

WT human myoblasts were collected from a cadaveric donor who did not have an RYR1-related disorder. Research using those cells was conducted in accordance with the Declaration of Helsinki and approved by the Ethics Committee of the Centre de recherche du CHU de Québec—Université Laval.

### 4.4. Cell Culture

HEK293T cells were cultured in DMEM medium (Wisent Inc., Saint-Jean-Baptiste, QC, Canada) supplemented with 10% FBS (Wisent Inc., Saint-Jean-Baptiste, QC, Canada), and 1% penicillin–streptomycin (Wisent Inc., Saint-Jean-Baptiste, QC, Canada). Cells were kept at 37 °C with 5% CO_2_ in a humidified incubator.

The human myoblasts were cultured in a homemade medium [43,44] made of the following components: DMEM medium, 20% FBS (Wisent Inc.), 16% medium 199 (Invitrogen™ Inc., Carlsbad, CA, USA), 1% penicillin–streptomycin (Wisent Inc.), Fetuin 25 μg/mL (Life Technologies, Carlsbad, CA, USA), hEGF 5 ng/mL (Life Technologies), bFGF 0.5 ng/mL (Life Technologies), insulin 5 μg/mL (Sigma-Aldrich Canada Inc., Oakville, ON, Canada, 91077C-1G), and Dex 0.2 μg/mL (Sigma-Aldrich Canada Inc.).

### 4.5. Ca^2+^ Leak Assay

RyR expressing HEK cell microsomes were prepared by centrifuging the cell lysates at 45,000× *g* for 30 min. Pellets were resuspended in lysis buffer (10 mM Tris maleate, pH 7.0) containing 300 mM sucrose. HEK cell microsomes (5 µg/mL) were diluted in a buffer (pH 7.2) containing 8 mM K-phosphocreatine, and 2 units/mL of creatine kinase, mixed with 3 µM Fluo-4 and added to multiple wells in a 96-well plate. The Ca^2+^ loading of the microsomes was initiated by adding 1 mM ATP. Fifty seconds after Ca^2+^ uptake, 3 μM thapsigargin was added to inhibit the Ca^2+^ reuptake by SERCA. Ca^2+^ leak was measured as the increase in intensity of the Fluo-4 signal (measured in a Tecan fluorescence plate reader). The Ca^2+^ leak was quantified as the increase in Fluo-4 signal/s for the first five seconds following the addition of the thapsigargin.

### 4.6. Transfection of HEK293T Cells

The day before transfection, 60,000 HEK293T cells were plated in a 24-well plate with 500 μL of culture medium per well. On the day of transfection, 1 μg of total DNA (500 ng of the plasmid coding for the prime editor and 500 ng of the plasmid coding for the epegRNA and the nsgRNA) was transfected with Lipofectamine 2000 (Invitrogen™ Inc., Carlsbad, CA, USA) following the manufacturer’s instruction. A control transfection with the eGFP plasmid was performed. The medium was changed 24 h later with 1 mL of fresh medium, and cells were maintained in the incubator for 48 h before genomic DNA extraction.

### 4.7. Plasmid Electroporation in Myoblasts

Electroporation was performed using the 10 μL Neon™ Transfection System (Invitrogen™) and its respective commercial kit following the supplier’s recommended protocol. In total, 100,000 myoblasts were electroporated for each condition with 1 µg of pCMV-PE2 (PE) and 1 µg of epegRNA-nsgRNA plasmid. Electroporations were performed at a voltage of 1100 V with two pulses of 20 ms. Myoblasts were then kept for 72 h in a 24-well culture plate with 1 mL of homemade medium changed after 24 h. For conditions using PE5, pCMV-PE2 was replaced with pEF1a-MLH1dn from AddGene (#174824). For PE5max, pCMV-PEmax-P2A-hMLH1dn from AddGene (#174824) was added instead pCMV-PE2. For PE6, pCMV-PE2 was replaced with pCMV-PE6a from AddGene (#207851). eGFP plasmid electroporation was also performed as a control.

### 4.8. PE mRNA In Vitro Transcription

The prime editor plasmid was first amplified by PCR to add optimized 5′ UTR regions and a poly-A tail in 3′ (Appendix A). PCR amplicons were purified with the PCR Products Purification Kit (EZ-10 Spin Column) (Bio Basic, Toronto, ON, Canada) and served as a template for subsequent in vitro transcription using the HiScribe T7 mRNA Kit with CleanCap Reagent AG (New England BioLabs Inc., Ipswich, MA, USA) with a complete replacement of UTP with N1-Methylpseudouridine-triphosphate (TriLink Biotechnologies Inc., San Diego, CA, USA). The reaction mixture was incubated at 37 °C for 3 h and 30 min. The volume of the reaction was then raised to 50 µL using nuclease-free water. DNase I (2 µL) was added, and the reaction mixture was left to incubate at 37 °C for 15 min. Transcribed mRNAs were purified using the Monarch RNA Clean-up Kit (500 µg) (New England BioLabs Inc., Ipswich, MA, USA), eluted in 1 mM Sodium Citrate pH 6.4, and quantified by BioDrop, Cambridge, UK. PE mRNAs were then stored at −80 °C.

### 4.9. RNA Electroporation in Myoblasts

PE mRNA (1 μg), epegRNA (4.6 μg) (chemically synthetized by IDT Inc., Coralville, IA, USA) and nsgRNA (1.8 μg) (chemically synthetized by IDT Inc., Coralville, IA, USA) were added to 100,000 human myoblasts and electroporated with the 10 μL Neon Transfection System (program: 1100 volts/20 ms/2 pulses). Cells were added to a 24-well culture plate with 500 μL of homemade medium per well. Control electroporations used eGFP mRNA (synthesized by IVT). The electroporation medium was changed after 24 h with 1 mL of fresh medium. DNA was extracted from these cells 48 h later.

### 4.10. Genomic DNA Preparation and PCR Amplification

HEK293T cells were directly detached from wells with the up-and-down pipetting of the medium and transferred in 1.5 mL Eppendorf tubes. Cells were centrifuged for 5 min at 8000 rpm. Cell pellets were washed with 500 μL of PBS and centrifuged again for 5 min at 8000 rpm. A mix containing 50 μL of DirectPCR Lysis Reagent (Viagen Biotech Inc., Los Angeles, CA, USA) and 0.5 μL of a proteinase K solution (20 mg/mL) for each sample was prepared. Then, 50.5 μL of this mixture was added to each cell pellet. Myoblasts were washed with 500 μL of PBS in their well. A mix of 100 μL of DirectPCR Lysis Reagent and 1 μL of a proteinase K solution (20 mg/mL) was added to each well. Samples were then incubated for 2 h at 56 °C followed by 45 min at 85 °C. Samples were then centrifugated at 13,500 rpm for 5 min. Next, 1 or 2 μL of each genomic DNA preparation (supernatant) was used for the PCR reaction. PCR temperature cycling was performed as follows: 30 s at 98 °C, then 35 cycles at 98 °C for 10 s, 58 °C for 20 s, and 72 °C for 30 s, and a final 5 min. elongation at 72 °C. Phusion™ High-Fidelity DNA polymerase from Thermo Scientific Inc. (Waltham, MA, USA) was used for all PCR reactions. Primers used for PCR are listed in Appendix A.

### 4.11. Sanger Sequencing

PCR amplicons were sent for Sanger sequencing to the sequencing platform of the CHU de Québec Research Center (https://sequences.ulaval.ca/murin/servseq.pageaccueil, accessed on 1 September 2023 to 15 December 2023). An internal primer (Appendix A) was used for polymerization with the BigDye™ Terminator v3.1. Sequences were analyzed with the EditR online program (https://moriaritylab.shinyapps.io/editr_v10/, accessed on 1 September 2023 to 20 December 2023) [45] to analyze editing efficiencies.

### 4.12. Statistical Analysis

Data were analyzed using the GraphPad PRISM 10.3.0 software package (Graph Pad Software Inc., La Jolla, CA, USA).

## 5. Conclusions

Our research explores an innovative and universal therapeutic approach for treating patients with RyR1-RM. We identified a protective mutation in the *RYR1* gene that successfully reduces pathological calcium leakage when introduced into cells carrying pathogenic *RYR1* mutations. Leveraging cutting-edge prime editing technology, we developed an RNA-based gene therapy to precisely insert this beneficial mutation into cultured human myoblasts, achieving an editing efficiency of 31%. This strategy represents a groundbreaking, once-in-a-lifetime treatment that could broadly apply to patients with a wide range of *RYR1* mutations. Given the over 700 known *RYR1* variants, this universal gene-editing approach marks a significant advancement in the field of genetic medicine.

## Figures and Tables

**Figure 1 ijms-26-02835-f001:**
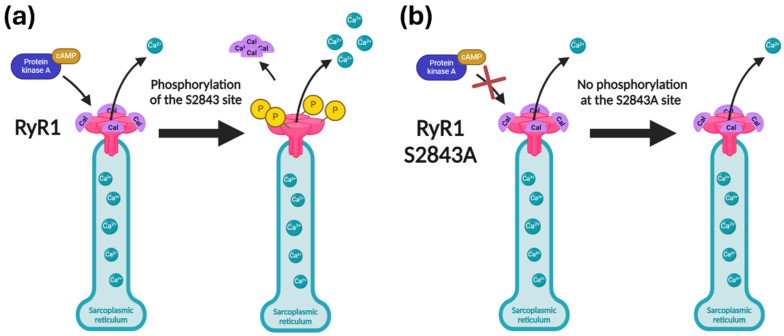
Impact of Ser2843 phosphorylation on RyR1 mechanism. (**a**) Calstabin in complex with a subunit of the RyR1 channel stabilizes its closed conformation. The action of cyclic AMP (cAMP)-activated protein kinase A (PKA) phosphorylates Ser2843 displaces calstabin and destabilizes the channel closed state, resulting in a leaky channel. (**b**) Ala2843 cannot be phosphorylated. Calstabin remains in complex with the channel, reducing the probability of channel opening.

**Figure 2 ijms-26-02835-f002:**
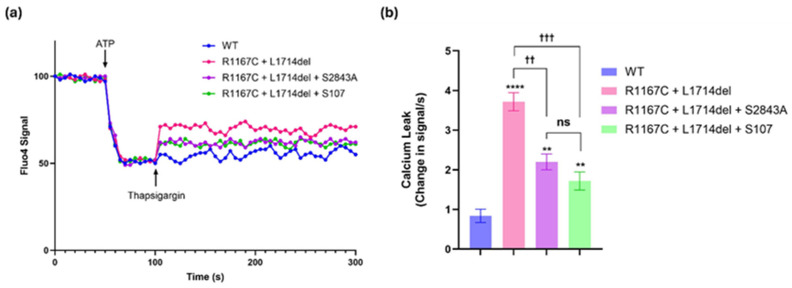
Calcium leak from RyR1 expressing HEK microsomes. (**a**) Ca^2+^ leak measured in microsomes from HEK cells expressing WT-RyR1, RyR1-R1667C + 1714del, RyR1-R1667C + L1714del + S2843A, and RYR1-R1667C + 1714del + S107. The graph of one trial is represented. The graphs of the four other trials are in Appendix A. (**b**) Bar graphs represent the quantification of the increase in Fluo-4 signals over the first 5 s after the addition of thapsigargin (change in signal/5 s = change/s). The 5 s time interval was selected as it provides the most precise measurement of the calcium ‘leak’, as reflected by the slope of the initial calcium release following thapsigargin treatment. Subsequently, cytosolic calcium levels stabilize, establishing a new equilibrium. Five trials of this experiment were performed. Thus, N = 5 for each group. ** represents a *p*-value < 0.01 vs. WT, and **** represents a *p*-value < 0.0001 vs. WT. †† represents a *p*-value < 0.01 vs. designated condition, and ††† represents a *p*-value < 0.001 vs. designated condition. ns represents a non-significative difference between (R1167C + L1714del + S2843A) and (R1167C + L1714del + S107).

**Figure 3 ijms-26-02835-f003:**
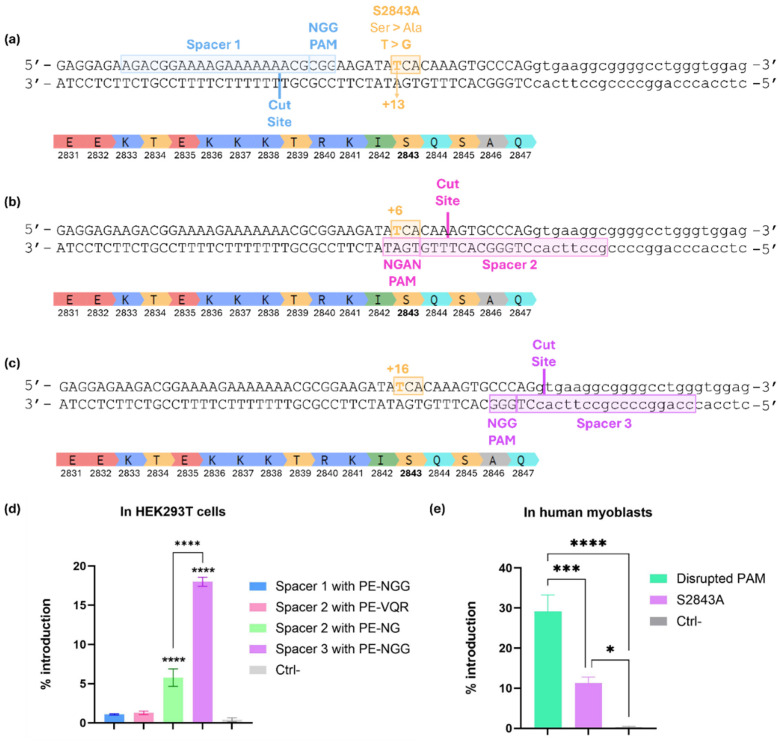
Introduction of the S2843A mutation in the *RYR1* gene by prime editing. (**a**–**c**) Sequence of the section of the *RYR1* gene containing the S2843 codon. The coding nucleotides are in capital letters, and those in the intron are in lowercase. Note that the sense strand is on the top of the figure. Three spacers were tested. The normal 2843 codon is TCA-coding for serine. To introduce the protective mutation S2843A, this codon is mutated to GCA-coding for alanine. (**a**) Spacer 1 uses an NGG PAM. The intended T>G substitution is at +12 from the cut site. The PAM (5′-CGG-3′) is in the sense strand, and it is the antisense strand that will be nicked by the SpCas9 nickase part of the PE. To introduce the protective mutation, a C must be inserted into the RTT sequence so that the reverse transcriptase synthesizes a new sense strand containing a 5′-GCA-3′ alanine sense codon. In this case, the PAM was not disrupted. (**b**) Spacer 2 uses an NGAN PAM. The intended T>G substitution is at +6 from the cut site. The PAM (5′-TGAT-3′) is in the antisense strand, and it is the sense strand that will be nicked by the SpCas9 nickase part of the PE. To introduce the protective mutation, a G must be inserted into the RTT sequence so that the reverse transcriptase synthesizes a new antisense strand containing a 5′-CGT-3′ alanine antisense codon. Note that the introduction of this mutation disrupts the PAM at the same time. It was tested with the PE-VQR and the PE-NG. (**c**) Spacer 3 uses an NGG PAM. The intended T>G substitution is at +16 from the cut site. The PAM (5′-GGG-3′) is in the antisense strand, and it is the sense strand that will be nicked by the SpCas9 nickase part of the PE. To introduce the protective mutation, a G must be inserted into the RTT sequence so that the reverse transcriptase synthesizes a new antisense strand containing a 5′-CGT-3′ alanine antisense codon. A mutation disrupting the PAM was also introduced (5′-GGG-3′ to 5′-GAG-3′). (**d**) Introduction of the S2843A mutation in the *RYR1* gene by prime editing in HEK293T cells. Transfection was carried out through the lipofection of plasmid DNA coding for the prime editing components (PE3). The mean of Spacer 1 includes the results of all the combinations of RTT-PBS lengths tested with Spacer 1 (see Appendix A). The mean of Spacer 2 includes the results of all the combinations of the RTT-PBS lengths tested with Spacer 2 (see Appendix A). Only one combination of the RTT-PBS length was performed with Spacer 3, and N = 3 replicates were performed. One-way ANOVA was used as a statistical test. **** represents a *p*-value < 0.0001. Efficiencies with Spacer 1 and Spacer 2 (with PE-VQR) were not significantly different from that of the negative control (Ctrl-). The efficiency of the epegRNA with Spacer 3 is significantly different from all other conditions with a *p*-value > 0.0001. (**e**) Introduction of the S2843A mutation in the RYR1 gene by prime editing in human myoblasts. The S2843A mutation is at +16 from the cut site. The PAM at +5 was also disrupted. Transfection was carried out through the electroporation of plasmid DNA coding for the prime editing components (PE3). N = 7 replicates were performed. One-way ANOVA was used as a statistical test. * represents a *p*-value < 0.05, *** represents a *p*-value < 0.001, and **** represents a *p*-value < 0.0001.

**Figure 4 ijms-26-02835-f004:**
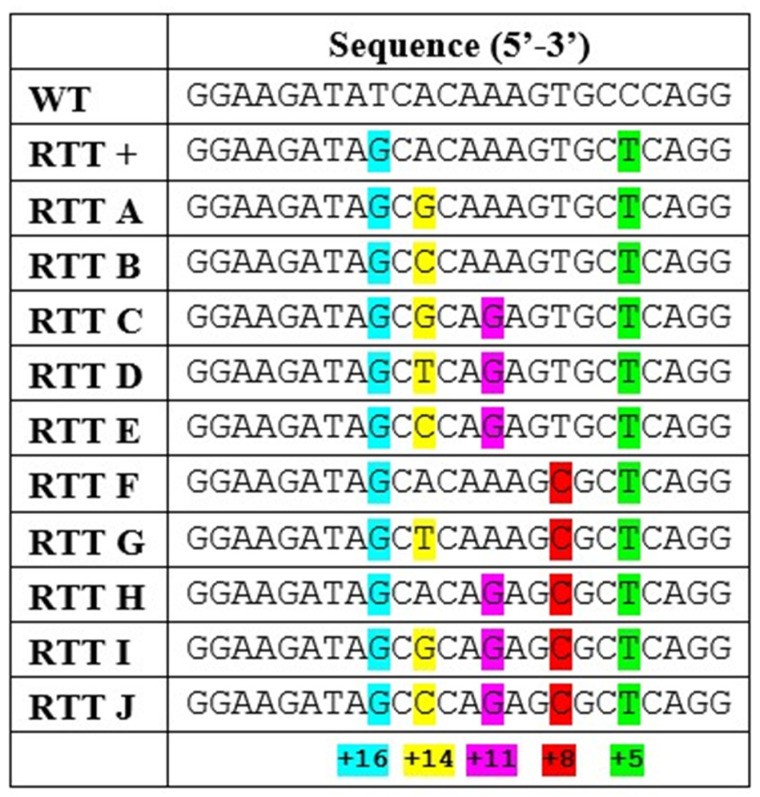
RTT sequence exhibiting the silent (same-sense) mutation combinations from the epegRNA targeting the introduction of the S2843A mutation. The S2843A mutation (+16) is in blue. The PAM silent mutation (+5) is in green. By counting from right to left, the position of the mutation from the cut site can be calculated. The +8 silent mutation is in red. The +11 silent mutation is in purple. The +14 silent mutation is in yellow. RTT+ represents the original RTT used in Figure 3d (Spacer 3) and 3e. The WT sequence is the one found in the human genome.

**Figure 5 ijms-26-02835-f005:**
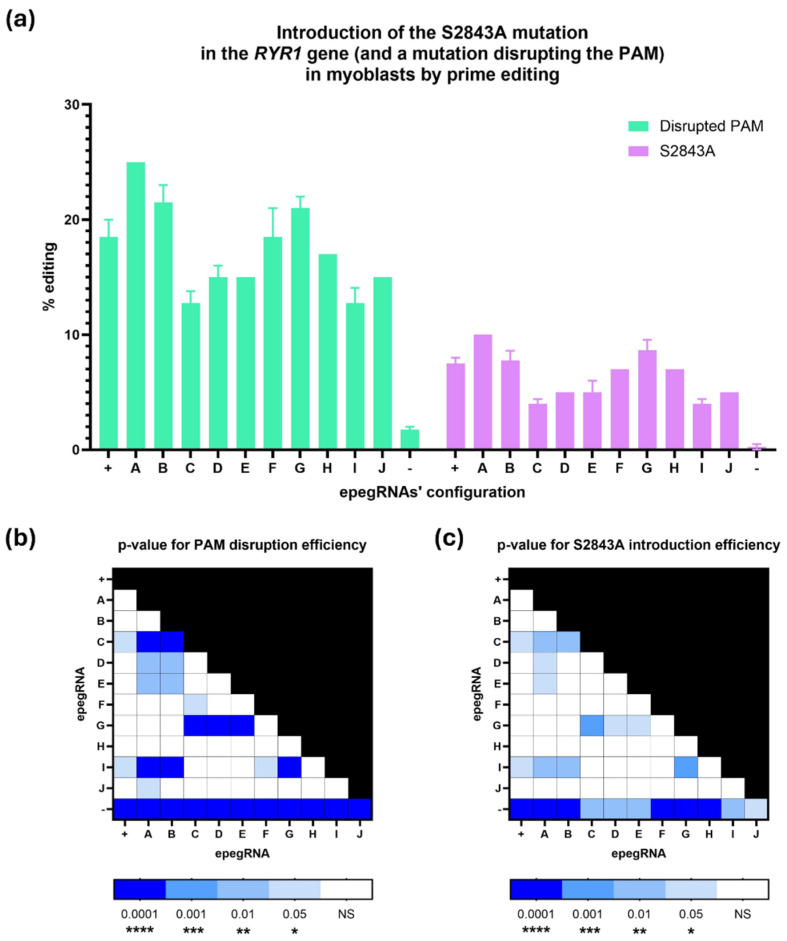
Effects of different combinations of silent mutations on the efficiency of prime editing to disrupt the PAM and to insert the S2843A mutation in the *RYR1* gene. (**a**) Prime editing efficiency to introduce the S2843A mutation in the RYR1 gene (+16 from the cut site) and to disrupt the PAM (+5 from the cut site) in wildtype myoblasts with the PE3 strategy. The + condition represents the epegRNA with only the +5 (PAM disruption) and +16 (S2843A) edits. Combinations of silent mutations for conditions A to J are specified in Figure 4. The—condition represents the negative control. The experiment was performed in biological duplicates. Tukey’s multiple comparison test was performed. *p*-values are represented in (**b**,**c**). (**b**) *p*-value of multiple comparisons of different epegRNA configurations for PAM disruption efficiency. (**c**) *p*-value of multiple comparisons of different epegRNA configurations for S2843A introduction efficiency. * represents a significant difference with a *p*-value of 0.05; **, of 0.01; ***, of 0.001; and ****, of 0.0001.

**Figure 6 ijms-26-02835-f006:**
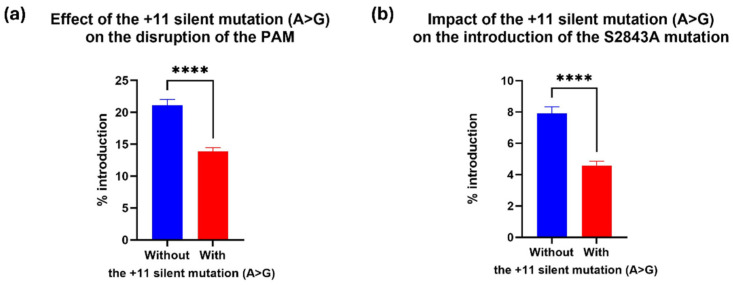
Impact of the +11 (A>G) silent mutation added in epegRNAs on prime editing efficiency. (**a**) Comparison of epegRNAs without the A>G mutation at +11 (+, A, B, F, and G) with those containing it (C, D, E, H, I, and J) to induce the C>T mutation in PAM at +5. (**b**) Comparison of epegRNAs containing the A>G mutation in +11 with those without it to induce the S2843A mutation (n = 10 for the group without the mutation in +11 and n = 12 for the group with the mutation in +11). An unpaired *t*-test was performed, and **** represents a significant difference with a *p*-value < 0.0001.

**Figure 7 ijms-26-02835-f007:**
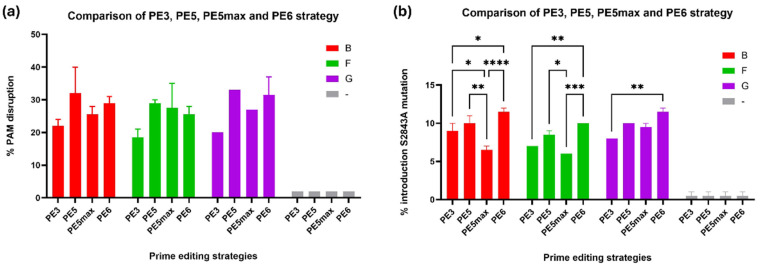
Comparison of different prime editing strategies (PE3, PE5, PE5max, and PE6) to disrupt the PAM and to introduce the S2843A mutation into the *RYR1* gene. (**a**) Efficiency of different prime editing strategies to disrupt the PAM in WT myoblasts by plasmid electroporation. (**b**) Efficiency of different prime editing strategies to introduce the S2843A mutation into the RYR1 gene in WT myoblasts by plasmid electroporation. EpegRNAs B, F, and G were used with the PE3, PE5, PE5max, and PE6 editing strategies. The—condition represents the negative control. The experiment was performed in biological duplicates. Tukey’s multiple comparison test was performed within each group (groups are the different epegRNAs: B, F, G, and -. * represents a significant difference with a *p*-value of 0.05; **, of 0.01; ***, of 0.001 and ****, of 0.0001. In (**a**), conditions within groups were not significantly different.

**Figure 8 ijms-26-02835-f008:**
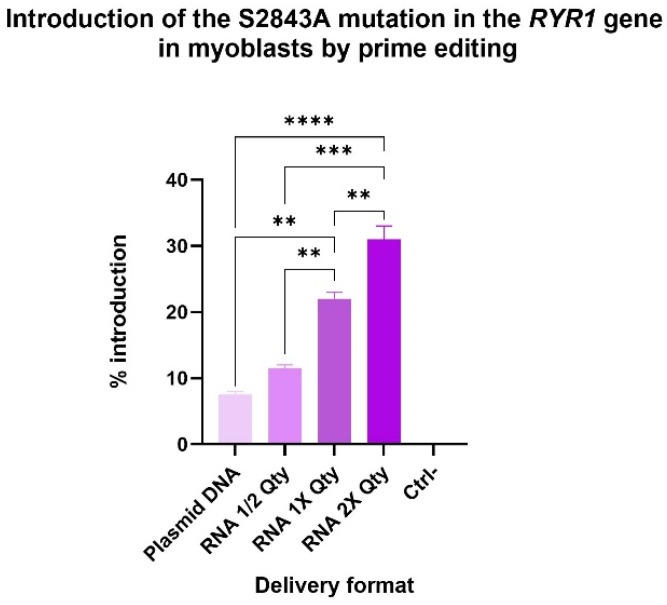
Introduction of the S2843A mutation in the *RYR1* gene by prime editing. The PE3 components were delivered by plasmid DNA or RNA by electroporation in WT myoblasts. RNA ½ Qty represents the following quantities: 0.5 µg PE mRNA, 2.3 µg of epegRNA, and 0.9 µg of nsgRNA. RNA 1X Qty represents the following quantities: 1 µg PE mRNA, 4.6 µg of epegRNA, and 1.8 µg of nsgRNA. RNA 2X Qty represents the following quantities: 2 µg PE mRNA, 9.2 µg of epegRNA, and 3.6 µg of nsgRNA. One-way ANOVA and Tukey’s multiple comparison test were used as a statistical test. All conditions are significantly different from the negative control (Ctrl-). ** represents a significant difference with a *p*-value of 0.01; ***, of 0.001 and ****, of 0.0001.

## Data Availability

The original contributions presented in this study are included in the article/Appendix A. Further inquiries can be directed to the corresponding author(s).

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
