# Peer review of "Universal Prime Editing Therapeutic Strategy for RyR1-Related Myopathies: A Protective Mutation Rescues Leaky RyR1 Channel"

_ijms, 2025, doi:10.3390/ijms26072835_

Round 1

Reviewer 1 Report

Comments and Suggestions for Authors

In this study, the authors tested and optimized the editing of the RyR1-S2843A mutation in vitro using prime editing, reporting an introduction rate of 31%, which significantly reduced Ca2+ leak caused by pathogenic mutations. Overall, this study suggests that the S2843A mutation has a protective effect, offering a potential universal therapeutic strategy for RyR1-related myopathies (RyR1-RM). However, the method used to assess editing efficiency is not reliable. To obtain an accurate assessment of editing efficiency, I suggest using targeted PCR sequencing with NGS. While EditR is suitable for quickly and intuitively evaluating prime editing efficiency, it may have limitations when dealing with complex edits or low signal-to-noise data.

Other comments:

Figure 1, the font size of some labels is too small and difficult to read.

Figure 2A and Figure S1 Trial 3 are duplicated; it is recommended to remove Trial 3 from Figure S1.

Reviewer 2 Report

Comments and Suggestions for Authors

This study explores a groundbreaking therapeutic strategy for RyR1-related myopathies (RyR1-RM), a group of genetic muscle disorders caused by numerous mutations in the RYR1 gene that disrupt calcium regulation within cells. The research zeroes in on the S2843A mutation, which, when introduced alongside disease-causing variants, stabilizes the RyR1 calcium channel and significantly reduces calcium leakage, as confirmed through a specialized assay. The team employed prime editing—a precise gene-editing technique—to incorporate this protective mutation into human myoblasts, achieving a 31% success rate via RNA electroporation in lab experiments. Given the vast number of RYR1 mutations (over 700 identified), this approach offers a potential universal solution, aiming to address the root cause of muscle dysfunction across all patients with defective RyR1 channels through a single, lifelong treatment. The findings pave the way for a transformative, one-size-fits-all intervention for RyR1-RM. This work underscores the power of innovative gene editing to tackle complex genetic disorders with broad applicability. I recommend its acceptance by the International Journal of Molecular Sciences. Here are some concerns for the authors:

Minor concerns:

1) In the 3rd paragraph of the Introduction section, the text style should be revised.

2) In the Discussion section, the limitations of the research should be replaced to the last of this section.

3) In section 4.5, blank space should be added between the number and the unit.

4) In the Methods section, the authors tend to start the sentence with number. The authors should revise such sentences with none start with number.

5) In section 4.10, the unit “RPM” should be revised as “rpm”.

6) The authors shoud add one Conclusion section to summary the work and the meaning of the manuscript.

7) The formation of all the references should be revised according to the request of the International Journal of Molecular Sciences.

Major concerns:

1)     I think the Introduction section should be revised. The authors detailed the mechanism using a Figure. However, Figure should not be read in an Introduction section. What is more, the writing of the Introduction section reads not scientific and logical enough especially the last paragraph and the 4th paragraph of the Introduction section read have no logical relationship with the last paragraph.

2)     It is very strange when reading the Discussion section. In general, the Discussion section should just need to introduce the work of the manuscript with one paragraph discussing the limitation of the work. However, the authors almost introduced their work with chaos sequence. Therefore, the authors are recommended to rewrite this section to make it read clear.

--------------------------------------------------------------------------------------------------------

This study explores a groundbreaking therapeutic strategy for RyR1-related myopathies (RyR1-RM), a group of genetic muscle disorders caused by numerous mutations in the RYR1 gene that disrupt calcium regulation within cells. The research zeroes in on the S2843A mutation, which, when introduced alongside disease-causing variants, stabilizes the RyR1 calcium channel and significantly reduces calcium leakage, as confirmed through a specialized assay. The team employed prime editing—a precise gene-editing technique—to incorporate this protective mutation into human myoblasts, achieving a 31% success rate via RNA electroporation in lab experiments. Given the vast number of RYR1 mutations (over 700 identified), this approach offers a potential universal solution, aiming to address the root cause of muscle dysfunction across all patients with defective RyR1 channels through a single, lifelong treatment. The findings pave the way for a transformative, one-size-fits-all intervention for RyR1-RM. This work underscores the power of innovative gene editing to tackle complex genetic disorders with broad applicability. I recommend its acceptance by the International Journal of Molecular Sciences. Here are some concerns for the authors:

1) In the 3rd paragraph of the Introduction section, the text style should be revised.

2) In the Discussion section, the limitations of the research should be replaced to the last of this section.

3) In section 4.5, blank space should be added between the number and the unit.

4) In the Methods section, the authors tend to start the sentence with number. The authors should revise such sentences with none start with number.

5) In section 4.10, the unit “RPM” should be revised as “rpm”.

6) The authors shoud add one Conclusion section to summary the work and the meaning of the manuscript.

Reviewer 3 Report

Comments and Suggestions for Authors

The manuscript presents the protective effects of the RyR1 - S2843A mutation. Using a calcium leak assay, the authors found that this mutation significantly reduced calcium leak when combined with pathogenic mutations. Prime editing was successfully used to introduce the S2843A mutation, achieving 31 % efficiency in human myoblasts by RNA electroporation. These findings provide the basis for a novel therapy that could benefit all patients with a leaky RyR1 channel leading to impaired intracellular calcium regulation and muscle dysfunction.

Suggestions:

  1. The manuscript lacks a conclusion section.
  2. References should be formatted according to the journal's guidelines.
  3. I encourage the authors to consider the possibility of introducing metal ions into nucleotides in future studies (in future manuscripts). Such an approach might allow the formation of biologically relevant complexes. The determination of the stability constants of these complexes by potentiometric methods would be a valuable achievement.

Round 2

Reviewer 1 Report

Comments and Suggestions for Authors

The authors have addressed my concerns, thanks.

Reviewer 2 Report

Comments and Suggestions for Authors

The authors have improved the quality of the manuscript through the 2nd round revision. The present version of the manuscript is recommended to be accepted by the International Journal of Molecular Sciences. Congratulations to the authors. However, the authors may need to take notice of the writing style of the title "4. Materials and Methods " and "Conclusion" according to the template of MDPI. I think these two errors may need to be revised in the following proof reading procedure.